# Integration of Genome-Wide Identification and Transcriptome Analysis of Class III Peroxidases in *Paeonia ostii*: Insight into Their Roles in Adventitious Roots, Heat Tolerance, and Petal Senescence

**DOI:** 10.3390/ijms252212122

**Published:** 2024-11-12

**Authors:** Li Li, Songlin He, Peidong Zhang, Dengpeng Li, Yinglong Song, Wenqian Shang, Weichao Liu, Zheng Wang

**Affiliations:** College of Landscape Architecture and Art, Henan Agricultural University, Zhengzhou 450002, China; lili199207@126.com (L.L.); hsl213@yeah.net (S.H.); huishan247@163.com (P.Z.); lidengpenga@outlook.com (D.L.); edward_song1989@163.com (Y.S.); qianqian656@163.com (W.S.)

**Keywords:** *Paeonia ostii*, class III peroxidase, adventitious root, heat resistance, flower senescence

## Abstract

As a plant-specific gene family, class III peroxidases (PODs) play an important role in plant growth, development, and stress responses. However, the POD gene family has not been systematically studied in *Paeonia ostii*. In this study, a total of 57 *PoPOD* genes were identified in the *P. ostii* genome. Subsequently, phylogenetic analysis and chromosome mapping revealed that *PoPODs* were classified into six subgroups and were unevenly distributed across five chromosomes. The gene structure and conserved motifs indicated the potential for functional divergence among the different subgroups. Meanwhile, four *PoPODs* were identified as tandem duplicated genes, with no evidence of segmental duplication. Using RNA-seq data from eight different tissues, multiple *PoPODs* exhibited enhanced expression in apical and adventitious roots (ARs). Next, RNA-seq data from AR development combined with trend analysis showed that *PoPOD30/34/43/46/47/57* are implicated in the formation of ARs in tree peony. Through WGCNA based on RNA-seq, two key genes, *PoPOD5/15*, might be involved in heat tolerance via ABA and MeJA signaling. In addition, real-time quantitative PCR (qRT-PCR) analysis indicated that *PoPOD23* may play an important role in flower senescence. These findings deepened our understanding of POD-mediated AR development, heat tolerance, and petal senescence in tree peony.

## 1. Introduction

As a large category of enzymes, peroxidases (EC 1.11.1.X) are bifunctional enzymes present in living organisms, which can generate reactive oxygen species (ROS) or oxidize various substrates with H_2_O_2_ to maintain intracellular ROS levels [1]. Based on their functional attributes and protein structure, peroxidases can be classified into two groups, namely hemoglobin peroxidases and non-hemoglobin peroxidases [2]. In addition to animal peroxidases, hemoglobin peroxidases are further subdivided into three categories. Class I peroxidases are distributed in microorganisms and plants, and their principal function is to eliminate excess H_2_O_2_ to prevent cellular damage [3]. Examples include catalase (CAT) and ascorbate peroxidase (APX). Class II peroxidases are found in fungi and are involved in lignin degradation [4]. Examples include manganese and lignin peroxidases. Class III peroxidases (PRXs; EC 1.11.1.7) are specific to plants and are often referred to by abbreviations such as POX, PER, or POD [1,5]. Recently, this class has been referred to as POD.

The role of Class III *POD* genes has been extensively demonstrated in numerous studies, highlighting their crucial involvement in plant growth and development as well as stress responses. For instance, *PODs* participate in various biological processes, including ROS scavenging, lignification [6], cell wall metabolism [7], auxin metabolism [8], wound healing [9], seed germination [10], pollen–pistil interactions [11], fruit ripening [12], and root development [13]. *Arabidopsis POD* genes (*AtPrx33* and *Atprx34*) were predominantly expressed in roots, and the overexpression of *AtPrx34* promoted root elongation [14]. In cucumber, *CsPrx73* overexpression enhanced the formation of AR and increased the elimination of ROS under waterlogged conditions [13]. Additionally, *PODs* play significant roles in the presence of abiotic stressors such as salt [15], drought [16], and cold stress [17], as well as with various biological stressors such as bacteria and fungi [18]. The rice *POD* gene (*OsPrx114*) was localized in the endoplasmic reticulum and plasma membrane. Compared with the control, *OsPrx114* overexpression in rice increased the activity of POD and CAT, reduced the accumulation of ROS, and improved drought tolerance [16]. In sweet potato, the overexpression of *IbPRX17* can maintain a high SOD and POD activity and proline content, reduce the accumulation of ROS, and enhance salt and drought tolerance [19]. Therefore, a genome-wide analysis of POD will facilitate our understanding of its role in biology, physiology, and stress responses. In recent years, the POD family has been identified in numerous species, such as *Arabidopsis thaliana* [20], *Zea mays* [21], *Populus trichocarpa* [7], *Pyrus bretschneideri* [22], and *Capsicum annuum* [23]. Nevertheless, to date, no previous systematic study of the POD family has been reported in tree peony.

The tree peony is a perennial woody plant belonging to *Paeonia*, Paeoniaceae, and serves as an important ornamental plant in China [24]. Simultaneously, it possesses ornamental and medicinal properties. Nevertheless, the tissue culture plantlets of tree peony exhibited poor rooting and low transplantation survival rates [25], severely limiting their large-scale production. Under natural circumstances, a single tree peony flower persisted for three to five days, and petal senescence significantly affected its ornamental period [26]. Exploring the response mechanism of petal senescence in tree peony holds great significance. During the production process, the hot weather in southern China caused leaf damage in tree peony [27], resulting in reduced planting effectiveness and a considerable decline in their ornamental value. Considering that the *PODs* play critical roles in root development, senescence, and stress responses, yet their function in tree peony remains largely unreported, a comprehensive identification and analysis of the PoPOD gene family is of considerable significance in revealing root development, flowering regulation, and heat tolerance in tree peony.

In this study, all members of the PoPOD gene family were identified utilizing *P. ostii* genome data [24]. Firstly, the physicochemical properties, phylogenetic relationships, chromosome localization, collinearity, conserved motifs, gene structure, and cis-elements were systematically analyzed. Secondly, the expression pattern of the PoPOD gene family in adventitious root (AR) formation was disclosed through trend analysis using transcriptome data of AR development in tree peony. Additionally, WGCNA analysis was conducted using heat-resistant transcriptome data to construct regulatory networks between transcription factors and *PoPODs*. Finally, qRT-PCR was employed to analyze the expression patterns of tree peony petals during flower senescence. The results of this study provide a foundation for further functional research on *PoPODs*.

## 2. Results

### 2.1. Identification and Characterization of PoPODs

Based on hmmsearch and BLASTP, a total of 57 *PODs* were identified in the *P. ostii* genome and designated as *PoPOD1-PoPOD57* according to their chromosomal locations (Table 1). These 57 PoPOD proteins encoded 183 (PoPOD7) to 436 (PoPOD23) amino acids, with molecular weights ranging from 19733.31 (PoPOD7) to 47449.47 (PoPOD23). The theoretical pI values ranged from 4.58 (PoPOD20) to 9.51 (PoPOD7), of which 25 PoPOD proteins were acidic amino acids (pI < 7) and 32 were basic amino acids (pI > 7). The grand average of hydropathicity of most PoPODs was lower than zero, indicating that the majority of PoPODs were hydrophilic proteins. Most PoPODs were predicted to localize to the chloroplast, while seven were located extracellularly, six were within the cytoplasm, four had dual localization (extracellular/vacuole, mitochondrion/vacuole, chloroplast/vacuole, and chloroplast/nucleus), two were in the nucleus and endoplasmic reticulum and one was in the plasma membrane (PoPOD32).

### 2.2. Phylogenetic Analysis of PoPODs

To elucidate the evolutionary relationship of PoPOD proteins, a Maximum Likelihood (ML) phylogenetic tree was constructed using the amino acid sequences of 57 PoPODs and 73 *Arabidopsis* PODs (Figure 1). Based on the classification of the *Arabidopsis* POD family, the POD proteins were divided into six subgroups (I-VI). Phylogenetic analysis revealed that the *PoPOD* genes exhibited an uneven distribution across each subgroup. For instance, subgroup VI contained the highest number of *PoPODs* (25), while subgroup IV contained none. The number of *Arabidopsis PODs* was higher than that of the tree peony in all other subgroups, except for subgroup VI.

### 2.3. Conserved Motif and Gene Structure Analysis of PoPOD Family

The conserved domains of PoPOD proteins were predicted using the MEME website. A total of ten motifs were identified, ranging from motif 1 to motif 10. As shown in Figure 2, a higher number (55) of consensus sequences were found in motif 4, while fewer numbers (46) were observed in motifs 5 and 9. Additionally, 54 *PoPODs*, except for *PoPOD10*/*38*/*56,* contained motif 3, and 54 *PoPOD* genes, except for *PoPOD4*/*15*/*23*, contained motif 4. Excluding *PoPOD7*/*10*/*21*/*38*, 53 *PoPODs* contained motif 7. Other PoPOD proteins contained motifs 3, 4, and 7, with the order consistently being 7, 3, and 4. It was worth noting that *PoPOD10*/*38* did not contain motifs 3 or 7, whereas *PoPOD21* contained two copies of motif 4 (Figure 2). The structural analysis of *PoPOD* genes revealed that the number of exons ranged from 2 to 12. Most had four exons and three introns, with 27 genes (47.36%) having three introns, and 15 genes (26.32%) having two introns. Notably, the *PoPOD23* gene had the highest number of exons and introns, with twelve exons and eleven introns, whereas *PoPOD17*/*24* only had two exons and one intron (Appendix A). Additionally, most of the *PoPODs* in subgroups I and V contained three introns, suggesting a similarity between these subgroups. Subgroups II, III, and VI displayed a diverse range of intron and exon numbers. These findings indicated the potential for functional divergence among the different subgroups.

### 2.4. Chromosomal Localization and Gene Collinearity of PoPODs

To gain deeper insights into the evolutionary relationship within the POD gene family, a gene collinearity analysis between *P. ostii* and *A. thaliana* was constructed. The results revealed that seven homologous gene pairs were found in *P. ostii* and *A. thaliana* (Figure 3A). Among these, two *PoPODs* were located on chromosomes 01 and 04, with one *PoPOD* gene present on each of the other chromosomes.

To reveal the chromosomal distributions of *PoPODs*, a graph of chromosomes was drawn using the TBtools software (v. 2.119). A total of 56 out of the 57 *PoPOD* genes were mapped on the five chromosomes of *P. ostii*. As shown in Figure 3B, the number of *PoPODs* varied across the chromosomes. For instance, chromosomes 02, 04, and 05 contained 13 *PoPODs*, chromosome 01 contained 12 *PoPODs*, while only 5 *PoPODs* were located on chromosome 03. Two tandem duplication events were observed on chromosomes 04 (*PoPOD36/PoPOD37*) and 05 (*PoPOD52/PoPOD53*). Notably, both duplicated genes belong to subgroup VI.

### 2.5. Analysis of Cis-Regulatory Elements in PoPODs

The 2.0 kb upstream regions of all *PoPODs* were extracted from the *P. ostii* genome sequence as putative promoters, and their cis-elements were predicted using the PlantCARE website. Fifty-seven *PoPODs* contained 7359 TATA-box and CAAT-box elements, which are defined as common elements (Figure 4A). Additionally, other cis-elements were categorized into the following six groups: light response, phytohormone, stress response, transcription factor binding site, and plant tissue (Figure 4B,C). A significant number of light response cis-acting elements (589) were discovered in the promoters of 57 *PoPODs*, averaging 10.33 elements per gene. Notably, *PoPOD57* had the highest number (20), while *PoPOD13/45* each contained only 2 elements (Figure 4B,C). The hormone response elements included 149 ABA, 142 MeJA, 56 GA, 38 SA, 32 auxin, 25 zein, and 21 GA-related elements (Figure 4B,C). Among these, there were also 48 *PoPODs* containing ABA response elements (ABRE), with *PoPOD47* having the highest number (14) (Figure 4B,C). Additionally, 38 *PoPODs* contained MeJA elements (CGtCA-motifs and TGACg-motifs); *PoPOD27/57* had the highest number (8) (Figure 4B,C). For stress responses, the elements primarily included anaerobic (129), mixed stress (27), and low-temperature response elements (26). Transcription factor binding elements were mainly MYB binding sites, with a total of 110 (Figure 4B,C). Plant tissue-specific elements were also identified in some *PoPODs* (Figure 4B,C). These results suggested that *PoPOD* genes in tree peony may play significant roles in hormone signaling and environmental stress responses.

### 2.6. Expression Profiling of PoPOD Genes in Different Organs

To investigate tissue-specific expression patterns of the *PoPOD* genes in tree peony, transcriptome sequencing data were obtained from eight different tissues, including apical, adventitious roots, stems, leaves, petals, stamens, pistils, and seeds. As shown in Figure 5, *PoPOD4*/*15*/*23*/*55* showed high expression levels in all tissues, while other *PoPODs*, such as *PoPOD9*/*34*/*40*/*47*/*54*, exhibited higher expression levels in the apical and ARs. In contrast, these five genes showed a relatively low expression in other tissues. Additionally, three genes (*PoPOD27*/*44*/*57*) were highly expressed in the stamens.

### 2.7. Expression Profiling and Trend Analysis of PoPOD Genes in an In Vitro AR Formation

In order to study the response of POD family genes within the in vitro ARs of tree peony, transcriptome data at three key time points during rooting were analyzed (Figure 6A). Initially, the genes were classified through trend analysis and divided into eight clusters according to their temporal expression trends. A total of 20 *PoPODs* were divided into eight clusters, with 6 *PoPODs* belonging to the eighth cluster and 4 *PoPODs* belonging to the fourth cluster (Figure 6B). The KEGG enrichment analysis revealed that genes within the eighth cluster were associated with the biosynthesis of secondary metabolites, starch and sucrose metabolism, MAPK signaling pathway-plant, endocytosis, and phenylpropanoid biosynthesis (Figure 6C). Subsequently, a total of eighty-six transcription factors including bHLH, C2H2, MYB, NAC, B3, and LBD were predicted using Plant TFDB, showing significant up-regulation on days five and ten (Figure 6D,E).

### 2.8. Expression Profiling and Co-Expression Network Analysis of PoPOD Genes in Response to High-Temperature Stress

To explore how *PoPODs* responded to heat resistance in tree peony, transcriptome sequencing data from high-temperature treatment (0 h, 2 h, 6 h, 12 h, 24 h) samples were analyzed. The results indicated that *PoPOD15*/*22*/*23*/*39*/*50* initially decreased followed by an increase when compared with the control group, whereas *PoPOD13*/*30*/*41*/*55* continued to decrease (Figure 7A). Furthermore, to elucidate the regulatory network involved in the *PoPODs*’ heat resistance response, a WGCNA co-expression network analysis was conducted. Initially, the WGCNA analysis identified nineteen modules, among which the blue module significantly correlated with heat resistance (Figure 7B). *PoPOD5*/*15* were found to belong to this module. Subsequently, a KEGG analysis for genes in the blue module revealed significant enrichments in pathways, including the biosynthesis of secondary metabolites, protein processing in the endoplasmic reticulum, plant-pathogen interactions, amino sugar and nucleotide sugar metabolism, ascorbate and aldarate metabolism, fructose and mannose metabolism, alpha-Linolenic acid metabolism, and glycerolipid metabolism (Figure 7C).

Furthermore, predictions of transcription factors within the blue module mainly included ERF, Trihelix, TCP, bHLH, and MYB (Figure 7D). A noteworthy observation was that the heatmap of the transcription factors primarily divided into two modules; those that were downregulated or upregulated after heat treatment (Figure 7E). Several *ERF* gene expressions were upregulated after treatment, while some *MYBs* and *GRASs* expression were shown to be downregulated following high-temperature treatment (Figure 7E). Finally, a regulatory network between transcription factors and *PoPOD5/15* was constructed based on their interactions in the blue module (Figure 7F).

### 2.9. qRT-PCR Analysis of PoPOD Genes in Flower Senescence

In order to investigate the role of *PoPODs* in the petal senescence of tree peony, five highly expressed *PoPODs* in petals were selected and analyzed using qRT-PCR following cut flower treatment at time points of one, two, three, four, and five days. During petal senescence, *PoPOD4/15/41/55* exhibited a downregulation followed by an upregulation trend, while *PoPOD23* showed an upregulation followed by a downregulation trend (Figure 8). The expression of *PoPOD4/15/41* peaked at day four, while *PoPOD23* reached its peak at day two. These results suggested that *PoPOD* genes were involved in petal senescence.

## 3. Discussion

PODs are plant-specific enzymes, which play a crucial role in plant growth and development, and stress responses. Given the significant biological function of *PODs*, the POD gene family has been extensively studied in model plants, crops [21], fruits [22], and trees [7]. Nevertheless, studies have yet to identify or characterize *POD* genes in tree peony. In this study, a total of 57 *PoPOD* genes were identified in the *P. ostii* genome, which is fewer than in *Arabidopsis thaliana* (73) [20], *Populus trichocarpa* (93) [7], and *Pyrus bretschneideri* (94) [22], but slightly more than in *Vitis vinifera* (47) [28]. This finding indicated that the POD gene family in *P. ostii* and *Vitis vinifera* has not expanded significantly compared with other plants. Subsequently, the physical and chemical properties, phylogenetic analysis, conserved motifs, gene structure, chromosome localization, collinearity analysis, cis-acting element analysis, and gene expression profile of *PoPODs* were comprehensively analyzed.

The PoPODs were categorized into six distinct subgroups (I-VI) through phylogenetic analysis, consistent with previous reports on *Glycine max* [29] and sugarcane [30]. Conserved motif analysis demonstrated that motif 4 was present in the majority of *PoPOD* genes. Meanwhile, some PoPOD proteins contained motifs 3, 4, and 7, and the order of these motifs remained consistent. These results indicated that PoPOD proteins possessed highly conserved attributes, similar to those reported in *Sorghum mosaic* [17] and *Pyrus bretschneideri* [22]. It is widely acknowledged that the evolution of multigene families is caused by the structural diversification of genes. The introns within *PoPODs* exhibited substantial variation. In rare instances, the number of introns reached a considerable amount, such as *PoPOD23*, which contained 11 introns. This finding indicated that *PoPOD* members have a certain diversity. The majority of *PoPODs* (73.68%) possessed only two or three introns, a pattern associated with stress-related genes having fewer introns, a phenomenon also observed in plants like *Vitis vinifera* [28], *Glycine max* [29], and *Pyrus bretschneideri* [22]. Additionally, most of the *PoPODs* in subgroups I and V contained three introns, while subgroups II, III, and VI had varying numbers of introns. These results suggested a structural conservation and diversification within and between *PoPOD* subgroups, supporting the potential functional diversity of *PoPODs*.

Whole-genome duplication, segmental duplication, and tandem duplication are the three principal modes of gene family expansion. Chromosome localization revealed that the *PoPOD* genes were widely distributed across five chromosomes, which was consistent with the distribution of *PODs* in *Arabidopsis thaliana* [20], *Populus trichocarpa* [7], and *Pyrus bretschneideri* [22]. Additionally, the results revealed that no fragment repetition events were identified in the tree peony genome, and only four genes were determined as tandem duplicates. However, a total of 78 segmental duplications and 25 tandem duplications were identified in *Nicotiana tabacum* [18], and more duplication events were also discovered in *Glycine max* [29], *Populus trichocarpa* [7], and maize [21], suggesting that the expansion of the POD gene family in tree peony differs from that of most plants, which might be related to the γ whole-genome duplication (WGD) event [24]. *PoPODs* were minimally influenced by segmental or tandem duplications.

In this study, eight distinct tissues were utilized to verify the tissue-specific expression patterns of 57 *PoPODs*. Here, some *PoPOD* genes were highly expressed in all tissues, while others exhibited low or no expression in certain tissues. In *Arabidopsis Thaliana*, some *POD* genes (*PRX39*/*40*/*57*) were highly expressed in the apical meristem and elongation zone boundaries and were implicated in root development [31]. Intriguingly, in this research, multiple *PoPOD* genes were observed to have high expression levels in the apical and adventitious roots, and a similar phenomenon was observed in maize [21] and rice [32]. Hence, it is hypothesized that these highly expressed *PoPOD* genes play a crucial role in root development. To further validate this hypothesis, RNA-seq data were used to analyze the expression level of the *PoPOD* genes during in vitro adventitious root development of tree peony. These findings indicated that cluster 8 encompassed the majority of *PoPOD* genes (*PoPOD30*/*34*/*43*/*46*/*47*/*57*), and this cluster was enriched in the biosynthesis of secondary metabolites and phenylpropanoid biosynthesis pathways. Additionally, *PoPOD30*/*34*/*57* contained meristem expression response elements, and *PoPOD46/57* contained auxin response elements. It is noteworthy that the *PoPOD43* homologous gene *AtPRX64* has been reported to be regulated by *AtMYB36* to maintain ROS balance and to participate in lateral root primordia formation [33], and the *PoPOD30* homologous gene, *AtIRR1*, might participate in adventitious root formation through ROS signals [34]. In conclusion, six *PoPODs* are implicated in the formation of ARs in tree peony.

An increasing number of studies have demonstrated that *PODs* are widely involved in various plant stress responses. Overexpression of the *POD* gene *TaPRX-2A* in wheat has been reported to activate antioxidant enzymes and the ABA pathway, reducing ROS accumulation and enhancing salt tolerance [15]. Several studies have shown that *PODs* are regulated by plant hormones such as SA, ABA, JA, and ET, and play central roles in plant defense mechanisms [1]. In this study, the blue module significantly associated with heat resistance was identified through WGCNA analysis, which contained two *PoPOD* genes (*PoPOD5/15*), and the regulatory network of their interaction with transcription factors was constructed. Notably, *PoPOD5/15* were observed to contain five ABA and 11 MeJA response elements. These results suggested that *PoPOD5/15* could be involved in the heat resistance of tree peony via ABA and MeJA signaling. Senescence represents the final stage of plant development, during which endogenous SA and ROS levels increase in senescent plant tissues, with ROS acting as an upstream regulator of the SA pathway. In chrysanthemum florets, POD, CAT, and other antioxidant enzymes initially increased and then decreased during the senescence process. This suggested that an imbalance between ROS production and antioxidant defense led to senescence-related damage in florets [35]. In *Iris versicolor* flowers, some antioxidant enzymes increased as the flowers opened and decreased significantly toward senescence [36]. In this study, the expression of *PoPOD23* exhibited an upregulation followed by a downregulation trend, and its promoter contained multiple hormone-responsive elements, such as SA and ABA. These results implied that *PoPOD23* may play a significant role in tree peony senescence.

## 4. Materials and Methods

### 4.1. Genome-Wide Identification of PoPOD Genes

The genome data of *P. ostii* and *A. thaliana* (TAIR 10) were obtained from the China National GeneBank DataBase (https://ftp.cngb.org/pub/CNSA/data5/CNP0003098/CNS0560369/CNA0050666/, accessed on 19 February 2023) [24] and the TAIR database (https://www.arabidopsis.org/, accessed on 12 August 2023) [37], respectively. The protein sequences of 57 POD proteins in *Arabidopsis* were downloaded from the TAIR database (https://www.arabidopsis.org/, accessed on 12 June 2024) [37]. These sequences were used as a query to perform the BLASTp search (*E* value < 10^−5^). On the other hand, the peroxidase domain (PF00141) was downloaded from the Pfam database (http://pfam.xfam.org/, accessed on 12 June 2024) [38], and we used the HMMER search to identify the possible *PoPODs*. Subsequently, the putative *PoPODs* were confirmed using CDD (https://www.ncbi.nlm.nih.gov/cdd, accessed on 14 June 2024) [39]. Finally, the physical and chemical properties and subcellular localization of the identified PoPOD proteins were calculated using the ExPASy-ProtParam tool (https://web.expasy.org/protparam/, accessed on 14 June 2024) [40] and the WoLF PSORT (https://wolfpsort.hgc.jp/, accessed on 14 June 2024), respectively.

### 4.2. Phylogenetic Analysis, Conserved Motif and Gene Structure

The Muscle method was used to construct the alignment of POD proteins from *Arabidopsis* and *P. ostii*. Then, the alignment results were subjected to IQTREE to perform a maximum likelihood (ML) phylogenetic tree with the bootstrap set to 5000 [41]. Using TBtools (v. 2.119) software, phylogenetic trees were created [42]. For conserved motifs, the amino acid sequences of PoPOD were submitted to the MEME online website (https://meme-suite.org/meme/tools/meme, accessed on 17 June 2024) [43] with a maximum of ten motifs, and images were generated by TBtools software [42]. The GFF3 file of the *P. ostii* was submitted to the TBtools software for the gene structure of *PoPODs* [42].

### 4.3. Chromosomal Localization, Gene Collinearity and CIS-Acting Element Analysis

*PoPOD* genes were mapped to the *P. ostii* chromosome using the TBtools software based on *P. ostii* genomic data [42]. For the chromosomal locations of *PoPOD* genes, we utilized the *P. ostii* genomic data and visualized the results using the TBtools software [42]. McscanX software was used to analyze the segment and tandem duplication events of *PoPOD* genes [44]. The intra-species synteny relationships of *P. ostii* genome *PODs* and the inter-species synteny relationships between *Arabidopsis* and *P. ostii* were identified by MCScanX software [44]. We used the TBtools to extract the *PoPODs* promoter sequences (upstream 2.0 kb) from the *P. ostii* genomic sequence data and then uploaded this to the PlantCARE database (http://bioinformatics.psb.ugent.be/webtools/plantcare/html/, accessed on 12 July 2024) for cis-acting element analysis [45]. TBtools was used for visualization [42].

### 4.4. Expression Profiles of PoPOD and Data Analysis

Based on the RNA-seq data from eight different tissues of the tree peony, including apical, stems, leaves, petals, stamens, pistils, seeds, (https://ftp.cngb.org/pub/CNSA/data5/CNP0003098/, accessed on 13 June 2024) [24] and adventitious roots, the tissue-specific expression of *PoPODs* was examined. Additionally, the expression of *PoPODs* at zero, three, five, and ten days during AR formation was examined in tree peony (the RNA-seq data were acquired from the corresponding author). The expression trends of genes during AR formation were analyzed using the OmicShare tools (https://www.omicshare.com/tools/, accessed on 15 September 2024). To identify the responses of *PoPODs* to high-temperature stress (0 h, 2 h, 6 h, 12 h, and 24 h), the transcriptome raw data were downloaded from the NCBI SRA (accession numbers: PRJNA1079236) [27]. The WGCNA was used to analyze the modules which significantly correlated with heat resistance. The WGCNA-shinyApp (https://github.com/ShawnWx2019/WGCNA-shinyApp, accessed on 15 September 2024) was used for analysis in TBtools [42]. The PlantTFDB website (http://planttfdb.gao-lab.org/, accessed on 15 September 2024) was used to predict the transcription factors. The KEGG enrichment analysis of the identified clusters and the blue module was analyzed using KEGG Enrichment in TBtools [42] and images were demonstrated by the OmicShare tools (https://www.omicshare.com/tools/, accessed on 15 September 2024). Co-expression network diagrams of transcription factors and PoPOD5/15 in the blue module were constructed using the OmicShare tools (https://www.omicshare.com/tools/, accessed on 16 September 2024).

### 4.5. Plant Materials, RNA Extraction and qRT-PCR

The tree peony (*Paeonia suffruticosa Andr*.) cultivar ‘Xue Ying Tao Hua’ originated from Luoyang, Henan province, China. The cultivar was chosen because of its upright flowers and early flowering, making it an important cut flower for tree peony. In this study, flower petal samples at one, two, three, four, and five days after the bottle insertion were collected for qRT-PCR analysis. The total RNA from the petal samples was isolated using a Quick RNA extraction Kit (0416-50, Huayueyang, Beijing, China). Using a kit with gDNA Clean (AG11728, AG, Hunan, China), 1 μg of total RNA was converted to cDNA. The qRT-PCR was performed using a SYBR^®^ Green Pro Taq HS Mix kit (AG11719, AG, Hunan, China). Then, the relative expression levels of *PoPODs* were calculated using the 2^−ΔΔt^ method for relative quantification. The primers were designed using TBtools and were listed in the Appendix A.

## 5. Conclusions

In this study, a genome-wide characterization of *PoPOD* genes was conducted for the first time in tree peony, with a particular focus on their role in AR development, petal senescence, and heat tolerance. Fifty-seven *PoPODs* distributed across five chromosomes were systematically identified in tree peony. Conserved motifs and gene structure indicated structural conservation and diversification within and between *PoPOD* subgroups, supporting the potential functional diversity of *PoPODs*. The chromosomal location and gene collinearity analyses revealed that *PoPODs* were minimally influenced by segmental or tandem duplications. RNA-seq data from eight different tissues showed that multiple *PoPOD* genes were observed to have high expression levels in the apical and ARs. Expression profiling combined with cis-elements analysis revealed that *PoPOD30*/*34*/*57* contained meristem expression response elements, and *PoPOD46*/*57* contained auxin response elements. WGCNA analysis of the transcriptome data revealed that *PoPOD5*/*15* were located in modules significantly associated with heat tolerance. The qRT-PCR analysis indicated that the expression of *PoPOD23* exhibited an upregulation followed by a downregulation trend and its promoter contained multiple hormone-responsive elements, such as SA and ABA. These findings offer valuable insights into the function of *PoPODs*, especially in root development, heat tolerance, and petal senescence, and establish a foundation for future research to validate their precise regulatory mechanisms.

## Figures and Tables

**Figure 1 ijms-25-12122-f001:**
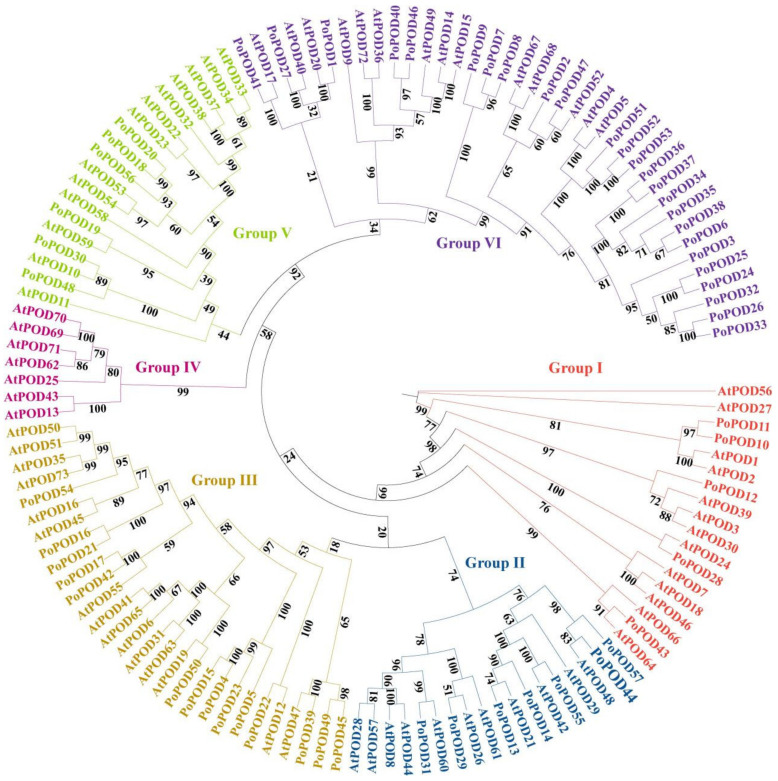
Phylogenetic relationship of *POD* genes between *Arabidopsis* and *P. ostii*. The phylogenetic tree was constructed by TBtools (v. 2.119) software using the Maximum Likelihood Method (5000 bootstrap). The numbers at nodes indicated bootstrap values per 5000 replicates as determined by the Maximum Likelihood Method. These proteins were divided into six subgroups (I–VI)), and were represented by bright red, dark blue, strong yellow, strong pink, strong green, and deep purple, respectively.

**Figure 2 ijms-25-12122-f002:**
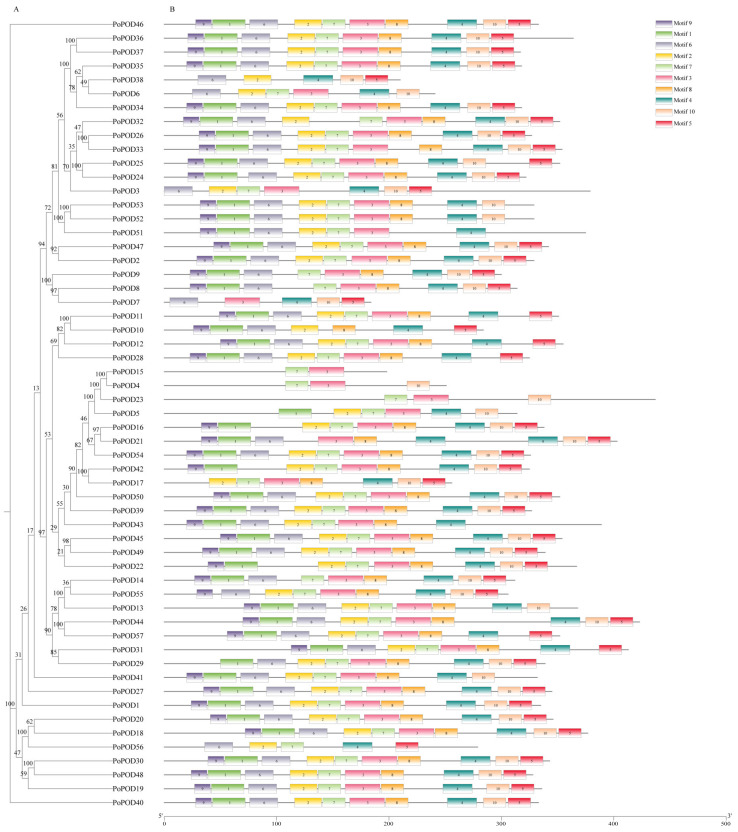
Phylogenetic evolutionary tree and motif composition of POD proteins in *P. ostii*. (**A**) The phylogenetic tree was constructed using full-length protein sequences of PoPOD proteins using the Maximum Likelihood Method (5000 bootstrap). (**B**) The amino acid motifs (numbered 1–10) in PoPOD proteins are displayed in ten colored boxes, and black lines indicate amino acid length.

**Figure 3 ijms-25-12122-f003:**
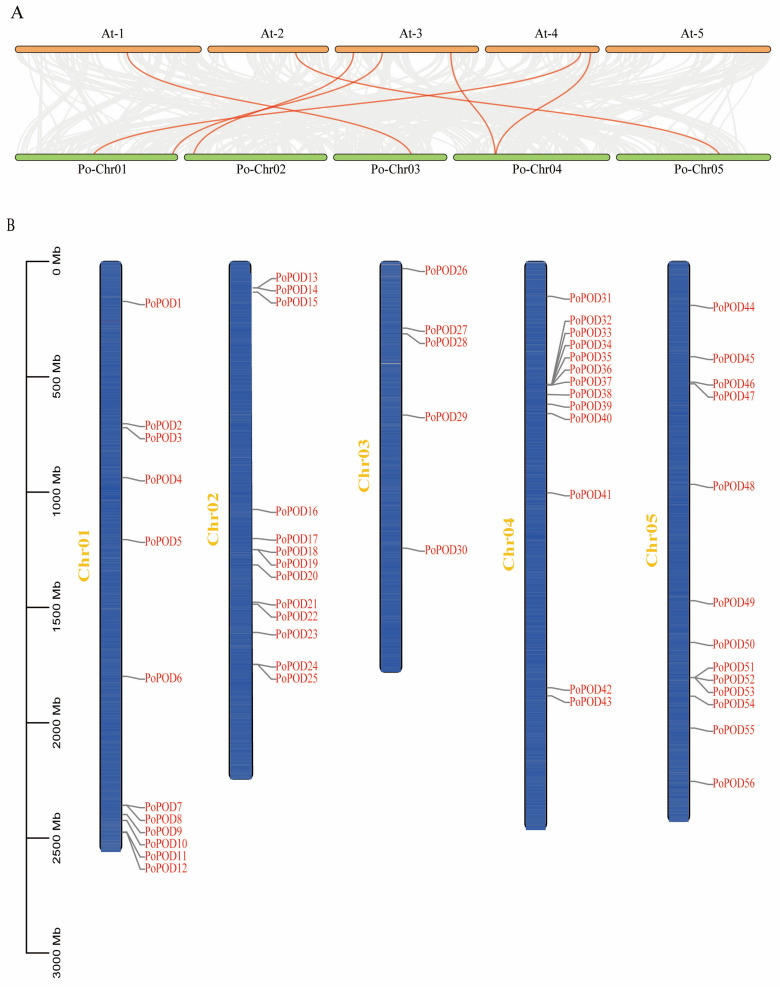
A gene collinearity analysis between *P. ostii* and *A. thaliana* and the chromosomal location of *PoPODs*. (**A**) A collinearity analysis of *PoPOD* genes between *P. ostii* and *A. thaliana* was constructed, and the red lines represented seven homologous gene pairs. (**B**) The blue bars represent the chromosomes of *P. ostii*; Chr 01–05 represent each corresponding chromosome, and the 0–3000 Mb scale represents chromosome length. A total of 56 out of the 57 *PoPOD* genes were located on the five chromosomes of *P. ostii* in addition to *PoPOD57*.

**Figure 4 ijms-25-12122-f004:**
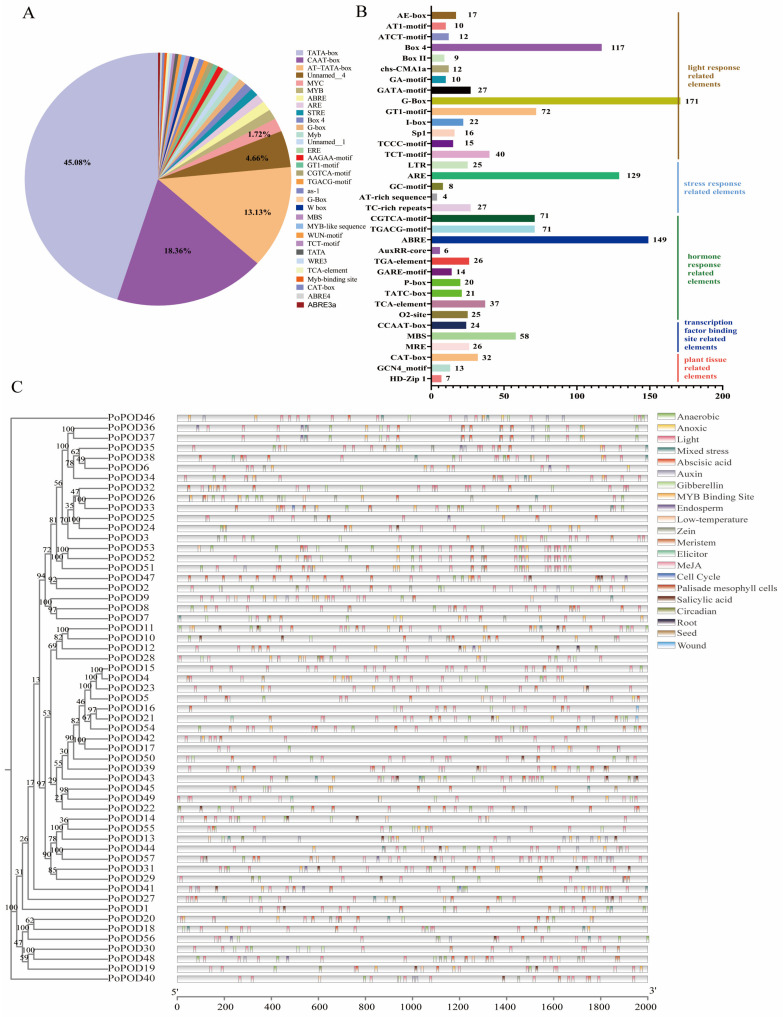
Cis-element analysis of *PoPOD* in *P. ostii*. (**A**) The proportion of cis-elements predicted in the promoters of *PoPOD*. (**B**) Numbers of the cis-elements involved in light response, phytohormone, stress response, transcription factor binding site, and plant tissue. (**C**) The distribution of the main cis-elements in *PoPOD* gene promoters.

**Figure 5 ijms-25-12122-f005:**
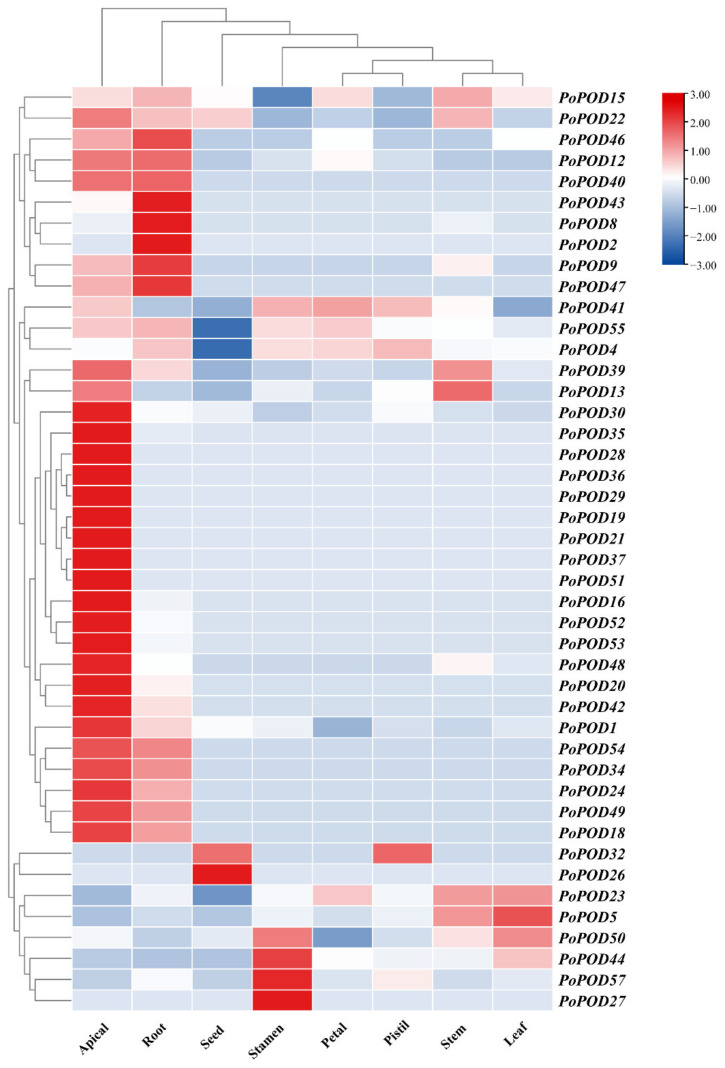
Expression profiles of tissue-specific *PoPOD* genes in different tissues. Heatmap of the expression in eight different tissues, including apical, root, seed, stamen, petal, pistil, stem, and leaf, by columns.

**Figure 6 ijms-25-12122-f006:**
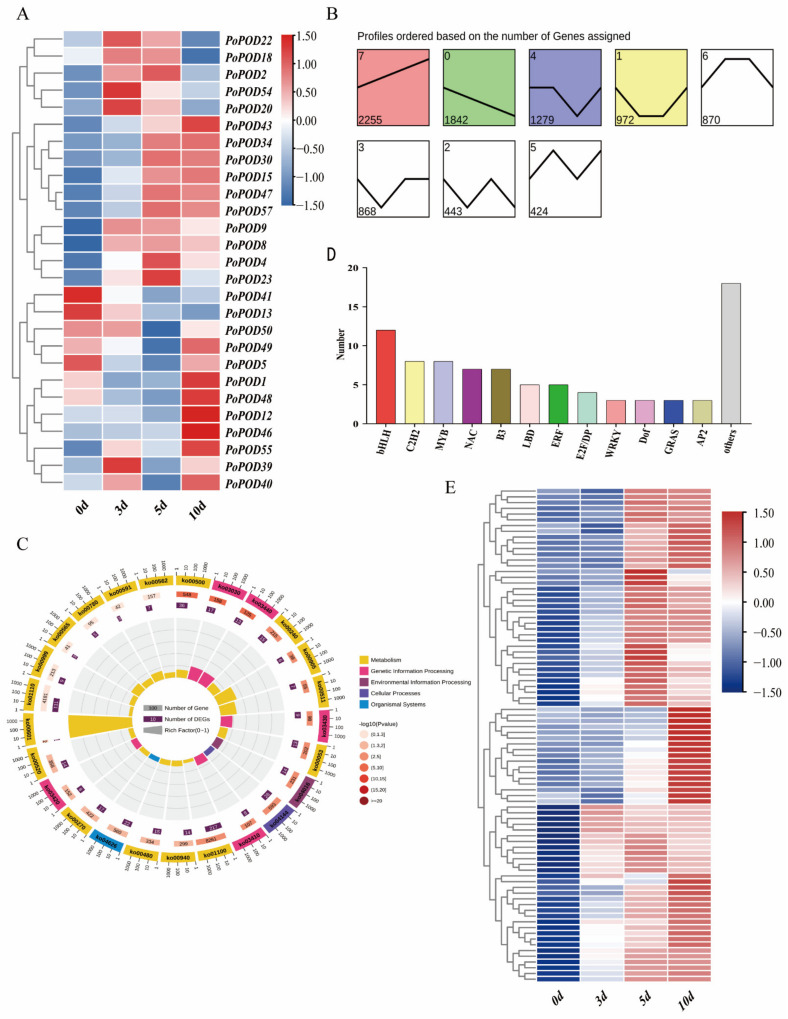
Transcriptome data analysis within the in vitro adventitious roots of the tree peony. (**A**) Expression profiles of *PoPOD* genes during in vitro adventitious roots of tree peony. (**B**) Trend analysis of the gene expression during in vitro adventitious roots of tree peony. The distinct colors in the background indicate different clusters. (**C**) KEGG enrichment analysis of cluster 8 genes. (**D**) Classification of transcription factors during in vitro adventitious roots in cluster 8. The distinct colors in the background indicate different transcription factors. (**E**) Heat map of transcription factor expression during in vitro adventitious roots in cluster 8.

**Figure 7 ijms-25-12122-f007:**
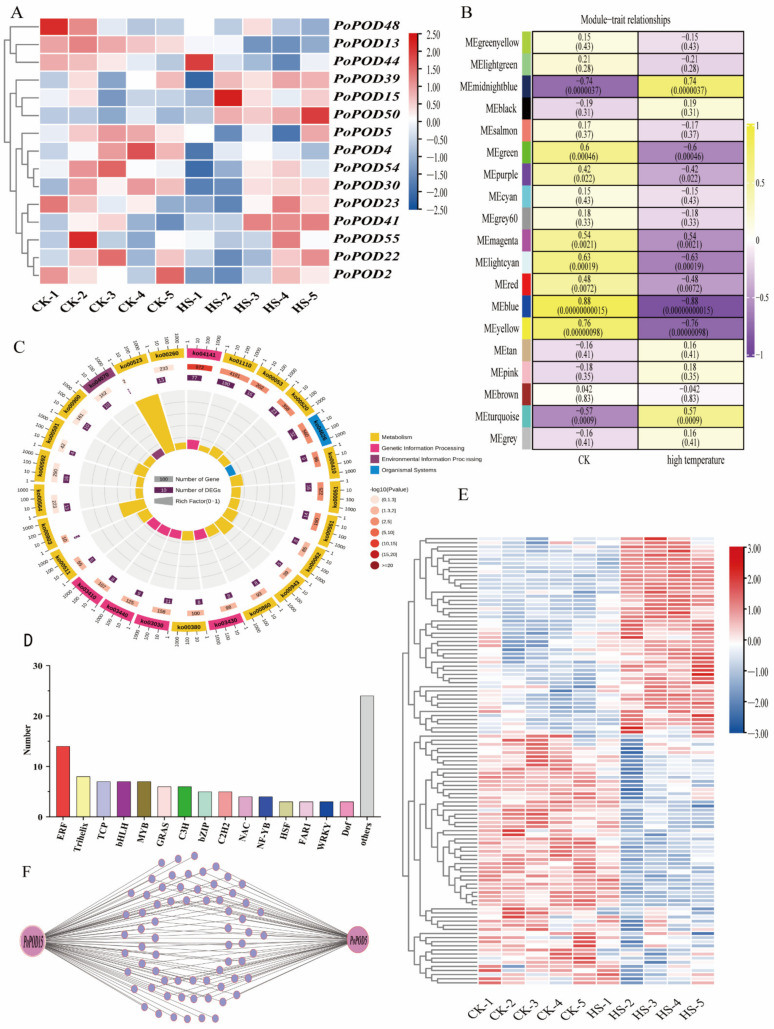
Transcriptome data analysis of heat resistance in the tree peony. (**A**) Expression profiles of *PoPOD* genes during high temperature treatment. (**B**) Heat map of the correlation between modules and traits. (**C**) KEGG enrichment analysis of blue module genes. (**D**) Classification of transcription factors in blue module. The distinct colors in the background indicate different transcription factors. (**E**) Heat map of transcription factor expression during high temperature treatment in blue module. (**F**) Co-expression network diagram of transcription factors and *PoPOD5/15* in blue module. The small blue circles indicate different transcription factors. The co-expression network was constructed using the OmicShare tools (https://www.omicshare.com/tools/, accessed on 16 September 2024).

**Figure 8 ijms-25-12122-f008:**
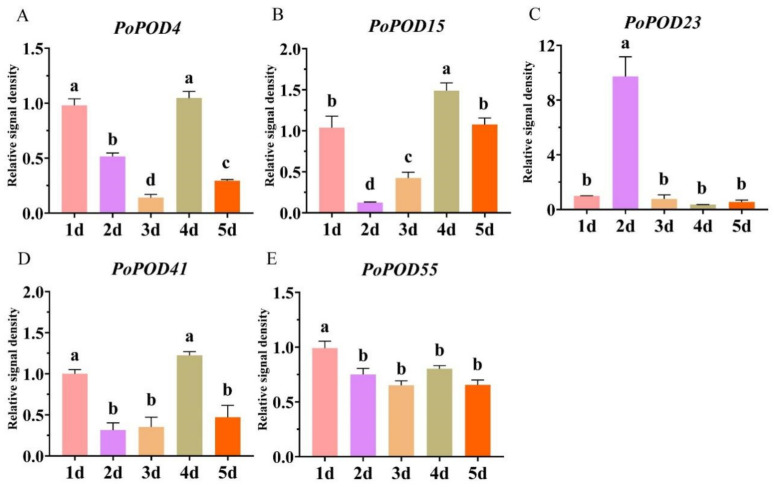
qRT-PCR analysis of five *PoPoD* genes in five different time periods. The mean ± SD from three biological replicates (*n* = 3) are shown. Different letters above the bars indicate a significant difference by Tukey’s test (*p* < 0.05, one-way ANOVA).

**Table 1 ijms-25-12122-t001:** The physicochemical parameters of POD proteins in the *P. ostii* genome.

Gene Name	Gene ID	AA	MW	pI	II	AI	GAH	Subcellular Localization
*PoPOD1*	*Pos.gene84304*	334	37,494.92	7.12	42.52	94.01	−0.094	Endoplasmic reticulum
*PoPOD2*	*Pos.gene13977*	328	34,971.5	8.87	38.66	80.03	−0.145	Chloroplast
*PoPOD3*	*Pos.gene40539*	378	40,274.49	6.19	40.08	73.57	−0.131	Nucleus
*PoPOD4*	*Pos.gene48550*	250	27,425.21	5.78	36.45	76.56	−0.367	Cytoplasm
*PoPOD5*	*Pos.gene2856*	313	34,272	6.33	46.66	81.34	−0.156	Chloroplast
*PoPOD6*	*Pos.gene41042*	240	25,949.54	9.43	33	83.75	−0.15	Chloroplast
*PoPOD7*	*Pos.gene20340*	183	19,733.31	9.51	20.14	72.08	−0.365	Nucleus
*PoPOD8*	*Pos.gene30160*	313	33,414.73	8.43	33.75	81.44	−0.092	Chloroplast
*PoPOD9*	*Pos.gene15411*	299	31,983.02	8.5	35.01	80.07	−0.1	Chloroplast
*PoPOD10*	*Pos.gene76296*	283	30,991.78	9.04	24.15	87.24	0.016	Extracellular
*PoPOD11*	*Pos.gene58720*	350	38,414.89	9.34	27.13	91.94	0.039	Chloroplast
*PoPOD12*	*Pos.gene36289*	354	38,758.27	8.08	47.06	88.73	−0.02	Cytoplasm
*PoPOD13*	*Pos.gene54483*	367	41,265.98	5.16	42.08	88.45	−0.269	Cytoplasm
*PoPOD14*	*Pos.gene50968*	311	35,284.08	6.37	40.74	78.68	−0.446	Chloroplast/Nucleus
*PoPOD15*	*Pos.gene75273*	198	21,817	6.65	36.79	80.81	−0.366	Cytoplasm
*PoPOD16*	*Pos.gene55938*	337	37,005.33	9.39	38.71	81.39	−0.188	Chloroplast
*PoPOD17*	*Pos.gene41371*	255	27,465.18	7.05	36.86	81.88	−0.16	Cytoplasm
*PoPOD18*	*Pos.gene39000*	376	40,087.66	4.9	38.96	81.28	−0.123	Chloroplast
*PoPOD19*	*Pos.gene39003*	335	35,654.32	6.2	34.58	87.58	0.015	Chloroplast
*PoPOD20*	*Pos.gene59972*	345	36,541.6	4.58	40.31	79.83	−0.1	Vacuole
*PoPOD21*	*Pos.gene34674*	402	45,305.81	9.31	40.42	74.55	−0.398	Mitochondrion/Vacuole
*PoPOD22*	*Pos.gene83645*	366	40,172.87	6.65	31.97	83.93	−0.13	Chloroplast
*PoPOD23*	*Pos.gene49612*	436	47,449.47	7.69	51.56	75.05	−0.376	Chloroplast
*PoPOD24*	*Pos.gene28436*	321	34,356.94	8.38	31.19	84.24	0.034	Chloroplast
*PoPOD25*	*Pos.gene51870*	351	38,243.43	8.99	40.61	85.36	−0.061	Chloroplast
*PoPOD26*	*Pos.gene66045*	326	35,222.76	5.35	38.25	85.89	0.051	Vacuole
*PoPOD27*	*Pos.gene27396*	344	37,363.27	4.64	35.36	91.28	0.041	Extracellular
*PoPOD28*	*Pos.gene54194*	324	35,276.15	8.95	44	87.56	−0.152	Extracellular/Vacuole
*PoPOD29*	*Pos.gene15815*	338	37,600.3	8.8	35.54	86.54	−0.224	Extracellular/Vacuole
*PoPOD30*	*Pos.gene4116*	342	37,835.89	6.44	29.31	81.55	−0.179	Chloroplast
*PoPOD31*	*Pos.gene81276*	412	45,147.99	8.73	38.97	91.38	0.085	Chloroplast
*PoPOD32*	*Pos.gene74750*	351	38,756.25	5.32	37.55	93.11	0.12	Chloroplast
*PoPOD33*	*Pos.gene74820*	353	38,242.51	6.8	34.75	87.08	0.143	Chloroplast
*PoPOD34*	*Pos.gene83499*	317	34,306.22	9.06	37.29	86.25	−0.096	Chloroplast
*PoPOD35*	*Pos.gene48117*	317	34,341.45	9.49	34.56	86.21	−0.117	Chloroplast
*PoPOD36*	*Pos.gene71590*	363	38,969.51	6.93	38.29	87.41	−0.021	Chloroplast
*PoPOD37*	*Pos.gene49860*	316	34,026.93	8.91	33.63	83.13	−0.083	Chloroplast
*PoPOD38*	*Pos.gene31703*	209	22,213.27	5.87	32.24	87.85	0.063	Extracellular
*PoPOD39*	*Pos.gene65483*	326	35,923.89	5.99	41.59	76.35	−0.145	Chloroplast
*PoPOD40*	*Pos.gene36467*	332	36,630.93	9.07	43.61	86.69	−0.214	Chloroplast/Vacuole
*PoPOD41*	*Pos.gene49996*	331	36,771.2	5.71	42.11	83.9	−0.183	Extracellular
*PoPOD42*	*Pos.gene23194*	324	35,145.02	4.9	36.64	84.91	0.007	Chloroplast
*PoPOD43*	*Pos.gene79357*	388	42,578.37	7.12	39.18	81.24	−0.256	Endoplasmic reticulum
*PoPOD44*	*Pos.gene54673*	422	47,085.99	5.04	68.37	69.38	−0.39	Chloroplast
*PoPOD45*	*Pos.gene41729*	353	38,929.48	7.57	39.86	79.58	−0.073	Chloroplast
*PoPOD46*	*Pos.gene76762*	332	36,370.44	8.93	42.65	87.32	−0.173	Extracellular
*PoPOD47*	*Pos.gene52501*	341	36,930.05	9.4	42.76	85.25	−0.065	Chloroplast
*PoPOD48*	*Pos.gene50909*	327	36,281.48	6.43	34.49	85.29	−0.103	Extracellular
*PoPOD49*	*Pos.gene45823*	338	36,567.49	8.06	42.76	81.92	−0.124	Extracellular
*PoPOD50*	*Pos.gene76474*	351	38,792.66	9.1	43.37	78.12	−0.238	Chloroplast
*PoPOD51*	*Pos.gene26113*	374	40,966.09	8.95	37.88	92.62	−0.009	Chloroplast
*PoPOD52*	*Pos.gene26110*	328	34,992.91	8.84	36.53	88.96	−0.023	Chloroplast
*PoPOD53*	*Pos.gene28373*	328	34,992.91	8.84	36.53	88.96	−0.023	Chloroplast
*PoPOD54*	*Pos.gene26703*	325	35,275.99	6.3	32.66	84.62	−0.066	Vacuole
*PoPOD55*	*Pos.gene82005*	305	34,282.11	6.8	40.91	85.31	−0.393	Cytoplasm
*PoPOD56*	*Pos.gene80919*	278	30,845.7	4.67	50.29	93.67	−0.088	Chloroplast
*PoPOD57*	*Pos.gene72397*	351	38,650.2	8.39	43.75	89.2	−0.093	Vacuole

AA, number of amino acid; MW, molecular weight; pI, theoretical pI; II, instability index; AI, aliphatic index; and GAH, grand average of hydropathicity.

## Data Availability

Data contained within the article.

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
