# Peer review of "Integration of Genome-Wide Identification and Transcriptome Analysis of Class III Peroxidases in Paeonia ostii: Insight into Their Roles in Adventitious Roots, Heat Tolerance, and Petal Senescence"

_ijms, 2024, doi:10.3390/ijms252212122_

Round 1

Reviewer 1 Report

Comments and Suggestions for Authors

Integration of Genome-wide Identification and Transcriptome of Class III Peroxidases (POD) in Paeonia ostii: Insight into Their Roles in Adventitious Roots, Heat Tolerance and Petal Senescence

I write you in regard to manuscript # International Journal of Molecular Sciences-3289316 entitled "Integration of Genome-wide Identification and Transcriptome of Class III Peroxidases (POD) in Paeonia ostii: Insight into Their Roles in Adventitious Roots, Heat Tolerance and Petal Senescence" which you submitted to the International Journal of Molecular Sciences.

Authors need to follow the following instructions to improve this manuscript

1)      In the title, the author mentioned POD. I think there is no need to use bracket and abbreviation. The title is very long. The authors should precise the title.

2)      Page 1, Abstract, line 13-14. Rewrite this sentence. Please avoid using “we”.

3)      Page 1, Abstract, line 23-25. Rewrite this sentence. Please avoid using “our”.

4)      Page 1, Abstract, line 26. Check the key words, comma and semicolon.

5)      Page 6, Para 2.3, line 118. Rewrite this sentence. Please avoid using “we”.

6)      Page 7, Para 2.4, line 140-141. Rewrite this sentence. Please avoid using “we”.

7)      Page 15, Para 3, line 262. Check the gap between trees and [7].

8)      Page 15, Para 3, line 263. In our study change to In this study

9)      Page 16, Para 3, line 298. RemoveWe concluded that”

10)   In this manuscript, there are many I, we, and our used. Please discard them and rewrite those sentences.

11)  References: should follow the journal guideline. I have seen somewhere full journal name and somewhere abbreviated form.

12)  English grammar should check carefully.

13)  Please check carefully before resubmission.

14)    Abstract should write on the basis of the best findings and their objectives.

15)    Conclusion should improve.

16)    In data, the author should add references if adopted from others.

 I recommend to improve the manuscript and resubmit.

Comments on the Quality of English Language

The English could be improved to more clearly express the research.

Author Response

Comments 1: In the title, the author mentioned POD. I think there is no need to use bracket and abbreviation. The title is very long. The authors should precise the title.

Response 1: Thank you for pointing this out. We have changed the title to the following “ Integration of Genome-wide Identification and Transcriptome of Class III Peroxidases in Paeonia ostii: Insight into Their Roles in Adventitious Roots, Heat Tolerance and Petal Senescence”. We have removed the bracket and abbreviation from the title in the revised text.

Comments 2: Page 1, Abstract, line 13-14. Rewrite this sentence. Please avoid using “we”.Response 2: Thank you for pointing this out. We agree with this comment. We have replaced “we identified a total of 57 PoPOD genes in the P. ostii genome” by “a total of 57 PoPOD genes were identified in the P. ostii genome” in the revised text (see Page 1, Abstract, lines 13-14 in the revised version).

Comments 3: Page 1, Abstract, line 23-25. Rewrite this sentence. Please avoid using “our”.

Response 3: Thank you for pointing this out. We agree with this comment. We have replaced “These findings deepened our understanding of POD-mediated AR development, heat tolerance, and petal senescence in tree peony” by “These findings deepened the understanding of POD-mediated AR development, heat tolerance, and petal senescence in tree peony” in the revised text (see Page 1, Abstract, lines 23-25 in the revised version).

Comments 4: Page 1, Abstract, line 26. Check the key words, comma and semicolon.

Response 4: Thank you for pointing this out. We agree with this comment. The keywords should be followed by a semicolon. We have revised this in the revised text (see Page 1, Abstract, line 26 in the revised version).

Comments 5: Page 6, Para 2.3, line 118. Rewrite this sentence. Please avoid using “we”.

Response 5: Thank you for pointing this out. We agree with this comment. We have replaced “We used the MEME website to predict conserved domains of PoPOD proteins” by “The conserved domains of PoPOD proteins were predicted using the MEME website” in the revised text (see Page 6, Para 2.3, line 119 in the revised version).

Comments 6: Page 7, Para 2.4, line 140-141. Rewrite this sentence. Please avoid using “we”.

Response 6: Thank you for pointing this out. We agree with this comment. We have replaced “To gain deeper insights into the evolutionary relationship within the POD gene family, we constructed a gene collinearity analysis between P. ostii and A. thaliana” by “To gain deeper insights into the evolutionary relationship within the POD gene family, a gene collinearity analysis between P. ostii and A. thaliana was constructed” in the revised text (see Page 7, Para 2.4, lines 142-143 in the revised version).

Comments 7: Page 15, Para 3, line 262. Check the gap between trees and [7].

Response 7: Thank you for pointing this out. We agree with this comment. We have added the gap between trees and [7] in the revised text (see Page 15, Para 3, lines 265-266 in the revised version).

Comments 8: Page 15, Para 3, line 263. In our study change to In this study

Response 8: Thank you for pointing this out. We agree with this comment. We have replaced “In our study” by “In this study” in the revised text (see Page 15, Para 3, line 267 in the revised version).

Comments 9: Page 16, Para 3, line 298. Remove “We concluded that”

Response 9: Thank you for pointing this out. We agree with this comment. We have removed “We concluded that” in the revised text (see Page 16, Para 3, line 302 in the revised version).

Comments 10: In this manuscript, there are many I, we, and our used. Please discard them and rewrite those sentences.

Response 10: Thank you for pointing this out. We agree with this comment. We have modified almost all of the “I, we, and our”. For example, we have replaced “We found that some PoPOD genes were highly expressed in all tissues, while others exhibited low or no expression in certain tissues” by “some PoPOD genes were highly expressed in all tissues, while others exhibited low or no expression in certain tissues” in the revised text (see Page 16, Para 3, lines 305-306 in the revised version). We have replaced “in our research” by “in this research” in the revised text (see Page 16, Para 3, lines 308-309 in the revised version). We have replaced “we hypothesized that these highly expressed POD genes play a crucial role in root development” by “it is hypothesized that these highly expressed POD genes play a crucial role in root development” in the revised text (see Page 16, Para 3, lines 311-312 in the revised version). We have replaced “To further validate our hypothesis, we employed RNA-seq data to analyze the expression level of the PoPOD genes during in vitro adventitious root development of tree peony” by “To further validate this hypothesis, RNA-seq data were used to analyze the expression level of the PoPOD genes during in vitro adventitious root development of tree peony” in the revised text (see Page 16, Para 3, lines 312-314 in the revised version). We have replaced “Our findings indicated” by “These findings indicated” in the revised text (see Page 16, Para 3, line 314 in the revised version). We have replaced “We hypothesized that” by “These results suggested” in the revised text (see Page 16, Para 3, line 332 in the revised version). We have replaced “we downloaded the peroxidase domain (PF00141) from the Pfam database” by “the peroxidase domain (PF00141) was downloaded from the Pfam database” in the revised text (see Page 17, Para 4.1, lines 353-354 in the revised version). We have replaced “we submitted the amino acid sequences of PoPOD to the MEME online website” by “the amino acid sequences of PoPOD were submitted to the MEME online website” in the revised text (see Page 17, Para 4.2, lines 365-366 in the revised version). We have replaced “we examined the tissue-specific expression of PoPODs” by “the tissue-specific expression of PoPODs was examined” in the revised text (see Page 17, Para 4.4, line 385 in the revised version). We have replaced “we examined PoPODs expression at 0, 3, 5 and 10 days during AR formation in tree peony” by “the expression of PoPODs at 0, 3, 5 and 10 days during AR formation in tree peony was examined” in the revised text (see Page 18, Para 4.4, lines 386-387 in the revised version). We have replaced “We used the WGCNA-shinyApp (https://github.com/ShawnWx2019/WGCNA-shinyApp) in TBtools for analysis” by “the WGCNA-shinyApp (https://github.com/ShawnWx2019/WGCNA-shinyApp) was used for analysis in TBtools” in the revised text (see Page 18, Para 4.4, lines 392-393 in the revised version). We have replaced “We systematically identified 57 PoPODs, distributed across five chromosomes, including two pairs of genes duplicated in tandem” by “57 PoPODs distributed across five chromosomes were systematically identified in tree peony” in the revised text (see Page 18, Para 5, line 415-416 in the revised version). We have replaced “We investigated their physicochemical properties, phylogenetic analyses, gene structures, and cis-element analyses, which indicated structural and functional diversity within the PoPOD gene family” by “Conserved motifs and gene structure indicated structural conservation and diversification within and between PoPOD subgroups, supporting the potential functional diversity of PoPODs” in the revised text (see Page 18, Para 5, line 416-418 in the revised version). We have replaced “Our findings offer” by “These findings offer” in the revised text (see Page 18, Para 5, line 428 in the revised version).

Comments 11: References: should follow the journal guideline. I have seen somewhere full journal name and somewhere abbreviated form.

Response 11: Thank you for pointing this out. We agree with this comment. We have modified the references according to the journal guideline in the revised text (see Page 19-20, References, lines 462, 466, 471, 475, 489, 491, 493-94, 498, 504, 508, 513, 526, 530-531, 531, 537, 541, 549, 551 and 554 in the revised version).

Comments 12: English grammar should check carefully.

Response 12: Thank you for pointing this out. We agree with this comment. We have checked the English grammar of the manuscript. An expert in good English edited the manuscript for proper English language, grammar, punctuation, spelling, and overall style.

Comments 13: Please check carefully before resubmission.

Response 13: Thank you for pointing this out. We agree with this comment. We have carefully checked before resubmission.

Comments 14: Abstract should write on the basis of the best findings and their objectives.

Response 14: Thank you for pointing this out. We agree with this comment. Based on your constructive suggestions, we have revised the Abstract. We have replaced “Subsequently, phylogenetic analysis, gene structure, conserved motifs, and chromosome mapping revealed that PoPODs were classified into six subgroups and were unevenly distributed across five chromosomes” by “Subsequently, phylogenetic analysis and chromosome mapping revealed that PoPODs were classified into six subgroups and were unevenly distributed across five chromosomes. Gene structure and conserved motifs indicated the potential for functional divergence among the different subgroups” in the revised text (see Page 1, Abstract, lines 14-17 in the revised version). We have replaced “RNA-seq data from AR development combined with trend analysis showed that PoPOD30/34/57 contained meristem expression response elements, and PoPOD46/57 contained auxin response elements” by “RNA-seq data from AR development combined with trend analysis showed that PoPOD30/34/43/46/47/57 are implicated in the formation of ARs in tree peony” in the revised text (see Page 1, Abstract, lines 20-21 in the revised version).

Comments 15: Conclusion should improve.

Response 15: Thank you for pointing this out. We agree with this comment. Based on your constructive suggestions, we have revised the Conclusion. We have replaced “57 PoPODs distributed across five chromosomes were systematically identified, including two pairs of genes duplicated in tandem. Their physicochemical properties, phylogenetic analyses, gene structure and cis-element analyses were investigated, which indicated structural and functional diversity within the PoPOD gene family. Expression profiling combined with trend analysis revealed that six PoPODs were clustered in cluster 8. WGCNA analysis of the transcriptome data revealed that PoPOD5/15 were located in modules significantly associated with heat tolerance. qRT-PCR analysis indicated that PoPOD23 exhibited a trend of increasing and then decreasing with the senescence process of the petals, peaking at day 2 of treatment” by “57 PoPODs distributed across five chromosomes were systematically identified in tree peony. Conserved motifs and gene structure indicated structural conservation and diversification within and between PoPOD subgroups, supporting the potential functional diversity of PoPODs. Chromosomal location and gene collinearity analysis revealed that PoPODs were minimally influenced by segmental or tandem duplications. RNA-seq data from eight different tissues showed that multiple PoPOD genes were observed to have high expression levels in the apical, ARs. Expression profiling combined with Cis-elements analysis revealed that PoPOD30/34/57 contained meristem expression response elements, and PoPOD46/57 contained auxin response elements. WGCNA analysis of the transcriptome data revealed that PoPOD5/15 were located in modules significantly associated with heat tolerance. qRT-PCR analysis indicated that the expression of PoPOD23 exhibited an upregulation followed by a downregulation trend and its promoter contained multiple hormone-responsive elements, such as SA and ABA” in the revised text (see Page 20, Conclusion, lines 415-427 in the revised version).

Comments 16: In data, the author should add references if adopted from others.

Response 16: Thank you for pointing this out. We agree with this comment. We have added references to data used in the manuscript in “Materials and methods”.

Reviewer 2 Report

Comments and Suggestions for Authors

Dear Authors,

Reviewer comments ijms-3289316

The manuscript entitled „Integration of genome-wide identification and transcriptome of class III peroxidases (POD) in Paeonia ostii: Insight into their roles in adventititous roots, heat tolerance and petal senescence“ represents a useful study combining genome-based data analysis of class III peroxidases (POD) from Paeonia ostii and expression analysis of selected POD genes isolated from Paeonia suffruticosa plant materials and analysed by qRT-PCR. I think that the manuscript provides a complex view on POD gene structure, phylogenetic analysis, cis-elements (promoter analysis) and expression analysis of POD genes in selected plant organs in tree peony species (P. ostii and P. suffruticosa) based on publicly available data and qRT-PCR expression analysis. I can recommend the manuscript for publication in International Journal of Molecular Sciences; however, I have some important comments on the present version of the manuscript which have to be addressed by the authors.

1/ Plant material used: It is evident that two peony species were used in the present study. The genome-based data use the P. ostii genomic data obtained from China National Genebank Database while the expression analysis using qRT-PCR was performed on P. suffruticosa plant materials. The authors should provide a brief commentary in Materials and methods while they used P. suffruticosa plants instead of P. ostii plants for qRT-PCR analysis, and what does the fact that two peony species were used in the study means for the interpretetation of the presented results.

2/ In the phylogenetic analysis, in Figure 1 legend, the statement about the numbers at nodes has to be added to the figure legend, i.e., „the numbers at nodes indicate bootstrap values per 5,000 replicates as determined by the Maximum likelihood method…“

3/ In Materials and methods, line 360, a relevant reference has to be added to „IQTREE software used to perform a maximum likelihood (ML) phylogenetic analysis.

4/ Some statements are unclear or confusing; their formulation has to be improved; e.g., Results, line 109: the authors stated that subgroup IV contained none PoPOD gene; it probably contains only AtPOD genes since both AtPOD and PoPOD genes were employed for the phylogenetic tree construction. Results, line 120: I do not understand the statment „ the higher number (55) of shared motifs was found in motif 4…“ – what does it mean „…a number of motifs in the motif…“ – this paradox has to eb clearly explained in the text.

5/ The abbreviations used in the text such as „ARs“ for „adventitous roots“ have to be explained either in the text when used for the first time or in a special abbreviations list attached to the manuscript.

6/ In Introduction, lines 39-40 when the alternative abbreviations used for peroxidases genes are listed the authors have to remove „Px“ and „Prx“ which are used for „peroxiredoxin“, not peroxidases. Peroxiredoxins represent a distinct protein family and cannot be confused with peroxidases.

7/ Fomal comments on the text related to English language and style:

Abstract, line 18: Replace the word „high“ with either „enhanced“ or „increased“ in the tstament „multiple PoPODs exhibite dan enhanced expression in apical and adventitious roots.“

Introduction, line 79: Modify the word form „secondly“ to „second“ in the statement: „Second, the expression pattern of the PoPOD gene family in adventitious root formation was disclosed…“

Results, line 94: Replace the word „less“ with „lower“ in the statement „The grand average of hydropathicity of most PoPODs were lower than 0…“

Results, Figure 3 legend, line 154: Remove the word „constructed“ in the statement „A gene colinearity analysis between P. ostii and A. thaliana and chromosomal location of PoPODs.“

Results, line 234: What is the difference between „was downregulated afetr high-temeprature treatment or downregulated afetr heat treatment“?? I think that one statement is sufficient!! i.e., „…was downregulated afetr heat treatment…“

Results, line 240, Figure 7 legend: Modify the word „treated“ to „treatment“ in the statement „during high temeprature treatment.“ In Figure 7 legend as well as in Materials and methods, it has to be also stated which software tools were used for the construction of the network diagram as presenetd in Figure 7F.

Discussion, line 265: Add the word „finding“ following „this“ in the statement „This finding indicates that the POD gene family in P. ostii and Vitis vinifera…“

Line 278: Moodify the words „structural diversity“ to „structural diversification“ in the statement „i tis widely acknowledged that the evolution of multigene families is caused by structural diversification of genes.“

Line 280. Add either the word „finding“ or „fact“ following „this“ in the statement: „This finding indicated that PoPOD members have a certain diversity.“

Line 296: Add a space between the reference „[7]“ and and „and“.

Materials and methods, line 343. Correct the typing error in the plant name „A. thaliana“ (not A. thanliana“) and add the word „respectively“ at the end of the statement: „The genome data of P. ostii and A. thaliana (TAIR 10) were obtained from China National GeneBank DataBase…..and the TAIR database…., respectively.“

Line 356: Add a space between the reference „[40]“ and „and“.

Line 360: Add a relevant reference on the software IQTREE used for the phylogenetic tree construction.

Line 405: Add a space between „table“ and „S1“ in „Supplementary Table S1“.

Conclusions, line 408: Modify the words „its role“ to „their roles“ in the statement: „In this study, genome-wide characterization of POD genes was conducted for the first time in tree peony, with a particular focus on their roles in AR development, petal senescence, and heat toelrance.“

Final recommendation: Accept after a minor revision.

Author Response

Comments 1: Plant material used: It is evident that two peony species were used in the present study. The genome-based data use the P. ostii genomic data obtained from China National Genebank Database while the expression analysis using qRT-PCR was performed on P. suffruticosa plant materials. The authors should provide a brief commentary in Materials and methods while they used P. suffruticosa plants instead of P. ostii plants for qRT-PCR analysis, and what does the fact that two peony species were used in the study means for the interpretetation of the presented results.

Response 1: Thank you for pointing this out. We agree with this comment. P. ostii was currently the best quality genome published for tree peony, so P. ostii genomic data were used in the present study. In qRT-PCR analysis, tree peony (Paeonia suffruticosa Andr.) cultivar 'Xue Ying Tao Hua' was used, which was often selected for cut flower generation, so this cultivar was used in the study. P. ostii is the parental species of varieties of modern cultivated tree peony (P. suffruticosa). Therefore, the use of two peony species in the study will not have a major impact on the results of the presented results. In order to cause unnecessary misunderstanding to the readers, we have explained the description in the Materials and Methods (see Page 20, Para 4.5, lines 401-403 in the revised version).

Comments 2: In the phylogenetic analysis, in Figure 1 legend, the statement about the numbers at nodes has to be added to the figure legend, i.e., „the numbers at nodes indicate bootstrap values per 5,000 replicates as determined by the Maximum likelihood method…“

Response 2: Thank you for pointing this out. We agree with this comment. We have added the sentence you mentioned to the legend of Figure 1(see Page 5, Para 2.2, lines 116-117 in the revised version).

Comments 3: In Materials and methods, line 360, a relevant reference has to be added to „IQTREE software used to perform a maximum likelihood (ML) phylogenetic analysis.

Response 3: Thank you for pointing this out. We agree with this comment. I have added a relevant reference ([41]) in the revised text (see Page 19, line 364 and Page 23, lines 544-545 in the revised version).

41 Nguyen, L. T.; Schmidt, H. A.; von Haeseler, A.; Minh, B. Q., IQ-TREE: a fast and effective stochastic algorithm for estimating maximum-likelihood phylogenies. Mol Biol Evol 2015, 32, (1), 268-74.

Comments 4: Some statements are unclear or confusing; their formulation has to be improved; e.g., Results, line 109: the authors stated that subgroup IV contained none PoPOD gene; it probably contains only AtPOD genes since both AtPOD and PoPOD genes were employed for the phylogenetic tree construction. Results, line 120: I do not understand the statment „ the higher number (55) of shared motifs was found in motif 4…“ – what does it mean „…a number of motifs in the motif…“ – this paradox has to eb clearly explained in the text.

Response 4: Thank you for pointing this out. We agree with this comment. Based on the classification of the Arabidopsis POD family, the POD proteins of AtPOD and PoPOD were divided into six subgroups (I-VI). As you comment, it probably contains only AtPOD genes since both AtPOD and PoPOD genes were employed for the phylogenetic tree construction. Thank you for your valuable suggestion. We apologize for this paradox. We have replaced “As shown in Figure 2, the higher number (55) of shared motifs was found in motif 4, while fewer motifs were observed in motifs 5 and 9 (46)” by “As shown in Figure 2, the higher number (55) consensus sequences were found in motif 4, while fewer numbers (46) were observed in motifs 5 and 9” in the revised text (see Page 5, Para 2.3, lines 120-122 in the revised version).

Comments 5: The abbreviations used in the text such as „ARs“ for „adventitous roots“ have to be explained either in the text when used for the first time or in a special abbreviations list attached to the manuscript.

Response 5: Thank you for pointing this out. We agree with this comment. We have replaced “adventitious root” by “adventitious root (AR)” in the revised text (see Page 2, Para 1, line 80 in the revised version).

Comments 6: In Introduction, lines 39-40 when the alternative abbreviations used for peroxidases genes are listed the authors have to remove „Px“ and „Prx“ which are used for „peroxiredoxin“, not peroxidases. Peroxiredoxins represent a distinct protein family and cannot be confused with peroxidases.

Response 6: Thank you for pointing this out. We agree with this comment. We have removed “Px, Prx”. The updated text now reads: “Class III peroxidases (PRXs; EC 1.11.1.7) are specific to plants and are often referred to by abbreviations like POX, PER, or POD” (see Page 1, Para 1, lines 38-40 in the revised version).

Comments 7: Fomal comments on the text related to English language and style:

Abstract, line 18: Replace the word „high“ with either „enhanced“ or „increased“ in the tstament „multiple PoPODs exhibite dan enhanced expression in apical and adventitious roots.“

Introduction, line 79: Modify the word form „secondly“ to „second“ in the statement: „Second, the expression pattern of the PoPOD gene family in adventitious root formation was disclosed…“

Results, line 94: Replace the word „less“ with „lower“ in the statement „The grand average of hydropathicity of most PoPODs were lower than 0…“

Results, Figure 3 legend, line 154: Remove the word „constructed“ in the statement „A gene colinearity analysis between P. ostii and A. thaliana and chromosomal location of PoPODs.“

Results, line 234: What is the difference between „was downregulated afetr high-temeprature treatment or downregulated afetr heat treatment“?? I think that one statement is sufficient!! i.e., „…was downregulated afetr heat treatment…“

Results, line 240, Figure 7 legend: Modify the word „treated“ to „treatment“ in the statement „during high temeprature treatment.“ In Figure 7 legend as well as in Materials and methods, it has to be also stated which software tools were used for the construction of the network diagram as presenetd in Figure 7F.

Discussion, line 265: Add the word „finding“ following „this“ in the statement „This finding indicates that the POD gene family in P. ostii and Vitis vinifera…“

Line 278: Moodify the words „structural diversity“ to „structural diversification“ in the statement „i tis widely acknowledged that the evolution of multigene families is caused by structural diversification of genes.“

Line 280. Add either the word „finding“ or „fact“ following „this“ in the statement: „This finding indicated that PoPOD members have a certain diversity.“

Line 296: Add a space between the reference „[7]“ and and „and“.

Materials and methods, line 343. Correct the typing error in the plant name „A. thaliana“ (not A. thanliana“) and add the word „respectively“ at the end of the statement: „The genome data of P. ostii and A. thaliana (TAIR 10) were obtained from China National GeneBank DataBase…..and the TAIR database…., respectively.“

Line 356: Add a space between the reference „[40]“ and „and“.

Line 360: Add a relevant reference on the software IQTREE used for the phylogenetic tree construction.

Line 405: Add a space between „table“ and „S1“ in „Supplementary Table S1“.

Conclusions, line 408: Modify the words „its role“ to „their roles“ in the statement: „In this study, genome-wide characterization of POD genes was conducted for the first time in tree peony, with a particular focus on their roles in AR development, petal senescence, and heat toelrance.“

Response 7: Thank you for pointing this out. We agree with these comments.

For Abstract, line 18, we have replaced “multiple PoPODs exhibited high expression in apical and adventitious roots (ARs)” by “multiple PoPODs exhibited enhanced expression in apical and adventitious roots (ARs)” in the revised text (see Abstract, line 18 in the revised version).

For Introduction, line 79, we have replaced “Secondly” to “Second” in the revised text (see Introduction, line 79 in the revised version).

For Results, line 94, we have replaced “The grand average of hydropathicity of most PoPODs were less than 0” to “The grand average of hydropathicity of most PoPODs were lower than 0” in the revised text (see Results, lines 93-94 in the revised version).

For Figure 3 legend, line 156, we have removed the word “constructed”. The updated text now reads: “A gene collinearity analysis between P. ostii and A. thaliana and chromosomal location of PoPODs”.

For Results, line 234, we have replaced “A noteworthy observation that heatmap of the transcription factors primarily divided into two modules was downregulated after high-temperature treatment or downregulated after heat treatment” to “A noteworthy observation that heatmap of the transcription factors primarily divided into two modules was downregulated or upregulated after heat treatment” in the revised text (see Results, lines 235-237 in the revised version).

For Results, line 240, Figure 7 legend, we have replaced “treated” to “treatment” in the revised text (see Results, lines 244, 247 in the revised version). We have added software tools in Figure 7 legend as well as in Materials and methods. For example, we have added “The co-expression network was constructed using the OmicShare tools (https://www.omicshare.com/tools/)” in the revised text (see Results, Figure 7 legend, lines 248-249 in the revised version). we have added “Co-expression network diagram of transcription factors and PoPOD5/15 in blue module was constructed using the OmicShare tools (https://www.omicshare.com/tools/)” in the revised text (see Materials and methods, lines 397-399 in the revised version).

For Discussion, line 265, we have added “finding” following “this”. The updated text now reads: “This finding indicates that the POD gene family in P. ostii and Vitis vinifera has not ex-panded significantly compared with other plants” (see Discussion, lines 269-271 in the revised version).

For Discussion, line 278, we have modified “structural diversity” to “structural diversification”. The updated text now reads: “It is widely acknowledged that the evolution of multigene families is caused by the structural diversification of genes” (see Discussion, line 282 in the revised version).

For Discussion, line 280, we have added “finding” following “this”. The updated text now reads: “This finding indicated that PoPOD members have a certain diversity” (see Discussion, lines 284-285 in the revised version).

For Discussion, line 296, we have added a space between the reference “[7]” and “and” in the revised text (see Discussion, line 300 in the revised version).

For Materials and methods, line 343, we have replaced “A. thanliana” to “A. thaliana” in the revised text (see Materials and methods, line 346 in the revised version). The updated text now reads: “The genome data of P. ostii and A. thaliana (TAIR 10) were obtained from China National GeneBank DataBase (https://ftp.cngb.org/pub/CNSA/data5/CNP0003098/CNS0560369/CNA0050666/, accessed on 19 February 2023) [24] and the TAIR database (https://www.arabidopsis.org/, accessed on 12 August 2023) [37], respectively”.

For Materials and methods, line 356, we have added a space between the reference “[40]” and “and” in the revised text (see Materials and methods, line 359 in the revised version).

For Materials and methods, line 360, I have added a relevant reference ([41]) in the revised text (see Materials and methods, line 364 and Page 23, lines 544-545 in the revised version).

41 Nguyen, L. T.; Schmidt, H. A.; von Haeseler, A.; Minh, B. Q., IQ-TREE: a fast and effective stochastic algorithm for estimating maximum-likelihood phylogenies. Mol Biol Evol 2015, 32, (1), 268-74.

For Materials and methods, line 405, we have added a space between “table” and “S1” in the revised text (see Materials and methods, lines 410-411 in the revised version). The updated text now reads: “In this study, genome-wide characterization of PoPOD genes was conducted for the first time in tree peony, with a particular focus on their role in AR development, petal senescence, and heat tolerance”.

For Conclusions, line 408, we have replaced “its role” to “their roles” in the revised text (see Conclusions, line 414 in the revised version).

Reviewer 3 Report

Comments and Suggestions for Authors

The paper performs a complete description in a huge family of genes, which are pivotal for stress response. From this perspective the research merits publication in IJMS. Nevertheless, I have some comments related to the presentation of the figures, which requires some improvements.

Table 1: Is not really clear. I would delete the second detimal position in the MW column and try to put all the information in a single row, with out line jumps. Another alternative is to put the page in horizontal to present the table in a clearer way.

Figure 4, 6 and 7: Some lettering cannot be read. Please enlarge. An alternative could be to present different pannels in different pages.

Figure 8: Please include in the legend a mention of the tests that has been performed (Tukey? Duncan?) 

Author Response

Comments 1: Table 1: Is not really clear. I would delete the second detimal position in the MW column and try to put all the information in a single row, with out line jumps. Another alternative is to put the page in horizontal to present the table in a clearer way.

Response 1: Thank you for pointing this out. We agree with this comment. We have increased the column width of Table 1, and it is manuscript upload system has compression causing Table 1 to be distorted.

Comments 2: Figure 4, 6 and 7: Some lettering cannot be read. Please enlarge. An alternative could be to present different pannels in different pages.

Response 2: Thank you for pointing this out. We agree with this comment. We have enlarged the lettering in Figures 4, 6 and 7 so that they can be read properly. Details of the revisions can be found in the revised text (see Figures 4, 6 and 7 in the revised version).

Comments 3: Figure 8: Please include in the legend a mention of the tests that has been performed (Tukey? Duncan?)

Response 3: Thank you for pointing this out. We agree with this comment. we have performed a mention of the tests (Tukey’s test) in the legend. We have replaced “Different letters above the bars indicate a significant difference (P < 0.05, one-way ANOVA)” to “Different letters above the bars indicate a significant difference by Tukey’s test (P < 0.05, one-way ANOVA)” in the revised text (see Results, lines 260-261 in the revised version).